# Algorithmic versus human surveillance leads to lower perceptions of autonomy and increased resistance
Rachel Schlund ✉ & Emily M. Zitek

Past research indicates that people tend to react adversely to surveillance, but does it matter if advanced technologies such as artificial intelligence conduct surveillance rather than humans? Across four experiments (Study 1, $N = 107$; Study 2, $N = 157$; Study 3, $N = 117$; Study 4, $N = 814$), we examined how participants reacted to monitoring and evaluation by human or algorithmic surveillance when recalling instances of surveillance from their lives (Study 1), generating ideas (Studies 2 and 3), or imagining working in a call center (Study 4). Our results revealed that participants subjected to algorithmic (v. human) surveillance perceived they had less autonomy (Studies 1, 3, and 4), criticized the surveillance more (Studies 1-3), performed worse (Studies 2 and 3), and reported greater intentions to resist (Studies 1 and 4). Framing the purpose of the algorithmic surveillance as developmental, and thus informational, as opposed to evaluative, mitigated the perception of decreased autonomy and level of resistance (Study 4).

The increasing prevalence of algorithmic surveillance affects people throughout organizations and society[1–3]. In algorithmic surveillance, artificial intelligence (AI) uses algorithms to track and analyze a wide range of individual behaviors and activities without the need for human discretion (e.g., vocal tone to detect stress levels, facial expressions to identify emotional states, written and verbal communication to understand interactions, etc.). Importantly, algorithmic surveillance often includes a monitoring component, which entails data collection and analysis, and an evaluation component, where decisions are derived from the analyzed data. For example, algorithmic surveillance is used in educational settings to detect and sanction cheating, in the workplace to track and evaluate the performance of workers, and in law enforcement to identify and apprehend individuals who engage in criminal behavior[1–4]. Although scholars from various disciplines have considered the impact of surveillance on people for decades[5–7], much of this work was conducted before the development of algorithmic surveillance. Despite the increasing use of algorithmic surveillance in lieu of prototypical forms of human surveillance, we know little about how this new form of surveillance impacts the individuals who are being monitored and evaluated.

Some anecdotes suggest that people may react adversely to algorithmic surveillance. For example, in educational settings, students seem opposed to the use of algorithmic surveillance compared to traditional surveillance by their teachers. Specifically, the increasing use of "fully algorithmic test monitoring," in which AI monitors and interprets students' behavior to detect and sanction cheating, has only been met with student complaints[1,8].

One student explained, "I feel like I can't take a test in my natural state anymore because they're watching for all these movements, and what I think is natural, they're going to flag."[1] In fact, the resistance from students subjected to fully algorithmic test monitoring has grown into full-fledged lawsuits and organized protests[1,8]. Is this a typical reaction to algorithmic surveillance?

We know from past work that people greatly value autonomy[9,10] and that surveillance can reduce perceived autonomy[11–13]. Crucially, because surveillance diminishes perceived autonomy, people may not always accept the surveillance; in some cases, individuals may even actively resist the surveillance[5,14,15]. If algorithmic surveillance reduces autonomy to a greater extent than human surveillance, people may prefer the latter. In this paper, we examine how algorithmic surveillance, compared to prototypical human surveillance, may threaten the autonomy of surveilled individuals, leading to more resistance. Decision-makers who are considering replacing prototypical human surveillance with algorithmic surveillance may focus on the potential benefits of this change[1,2]. However, if algorithmic surveillance reduces perceived autonomy, there could also be major negative consequences if people are unwilling to comply and instead actively resist its use.

When an individual is under surveillance, the monitoring and evaluation process may lead them to believe that they need to behave in specific ways to obtain rewards or avoid punishments (e.g., students feel that they cannot move while taking a test; otherwise, they might get penalized for cheating). This reduces their perceived autonomy[16,17] and may consequently change their other attitudes and behaviors[12]. Although subjecting

ILR School, Department of Organizational Behavior, Cornell University, Ithaca, NY, USA. ✉e-mail: rjs542@cornell.edu

individuals to surveillance can reduce their perceived autonomy[11,13], algorithmic surveillance may decrease individuals' perceived autonomy even further than prototypical human surveillance.

Supporting this idea, we know that individuals have lay theories about how algorithmic decision-making works and differs from human decision-making, even though those lay theories are not always accurate[18]. Specifically, individuals tend to think that algorithmic-driven systems cannot process information holistically, contextually, or thoroughly[19]. As Candelon et al. (2021) describe, "It's difficult for people to accept that machines can process highly contextual situations" (p. 107)[20]. Individuals perceive that algorithmic judgment quantifies behavior out of context, cannot take an individual's unique characteristics and circumstances into account[21], and consequently cannot "manage" complex tasks[22]. Thus, individuals may also perceive that algorithmic surveillance will fail to accurately evaluate their behavior or exercise discretion when appropriate. Consequently, people may feel especially constrained in how they can behave because they may worry that the algorithm will misinterpret what they are doing unless they behave in specific ways. Turning back to the example of "algorithmic test monitoring," one student described how she constrained her behavior, saying, "I try to become like a mannequin during tests now" out of her fear of being misinterpreted by the algorithmic surveillance[23].

Conversely, when it comes to human decision-making, people tend to hold an "illusionary understanding."[24] That is, people tend to believe they understand the processes behind human decision-making (e.g., how and why other humans make certain decisions) to a greater extent than they actually do. Importantly, people also tend to overestimate the extent to which other human decision-makers understand their internal states (such as the intentions guiding their behaviors), referred to as the "illusion of transparency."[25,26] Thus, people may believe that humans who are surveilling them will understand them better and, therefore, will be more likely to exercise discretion when necessary and will correctly interpret their behaviors. Therefore, people may not feel as constrained in how they can behave when subjected to human (vs. algorithmic) surveillance. Consequently, we hypothesize that algorithmic (as opposed to human) surveillance leads to lower perceived autonomy.

Importantly, research on psychological reactance and compensatory control has shown that when individuals perceive a loss of autonomy or control, such as when they are put under surveillance, they may become motivated to restore control[5,9,27–29]. To restore it, people might resist the prescribed behavior, engage in the proscribed behavior[30], or criticize the perceived source of the threat[31,32], which we term *resistance behaviors*. For example, past work has shown that when people perceive motivational speakers as controlling (as opposed to autonomy-supportive), they are more likely to engage in acts of resistance (behavior that is opposite of the behavior that the speaker is encouraging) to restore their autonomy[32]. Applying this to the current context, the more a surveillance method reduces autonomy, the more we can expect that the surveilled individuals will engage in resistance behaviors to restore their autonomy. Hence, we hypothesize that algorithmic (as opposed to human) surveillance leads individuals to engage in more resistance behaviors because of their reduced sense of autonomy.

Algorithmic surveillance, however, does not always have to lead to perceived autonomy reductions in comparison to human surveillance. Although surveillance is often conducted to monitor and evaluate the surveilled individuals[14], surveillance can also be conducted for developmental or informational reasons[33]. Individuals may be more accepting of algorithmic surveillance in these contexts. People generally feel less autonomy in evaluative contexts than in developmental contexts[13,17,33]. In evaluative contexts, people need to be careful to produce a positive evaluation. Conversely, in developmental contexts, people receive feedback or other information to assist them in their self-determined endeavors[11,13,17,34], which is less constraining. For instance, imagine a developmental context where a smartwatch is used to provide an individual with feedback to *assist* them in their self-determined endeavors, such as meeting their goal of moving a certain distance per day. In this context, if the smartwatch misevaluates their behavior (e.g., perhaps it failed to use discretion when they biked instead of

walked and thus did not factor this physical activity into their count), they can contextualize this feedback for themselves. They also might like that no one else sees their results (and therefore cannot judge them)[35]. However, when the same smartwatch is used in an evaluative context to *ensure* that they move a certain distance per day (e.g., to qualify for a deduction on their insurance premium), they may feel less autonomous in this context. In this case, if the smartwatch misevaluates their behavior, they might feel stuck with the erroneous result, which reflects poorly upon them. Accordingly, they may feel constrained in the type of activity they can engage in out of concern that the algorithm will fail to utilize discretion and consequently misevaluate their behavior. Accordingly, reframing the purpose of the monitoring as developmental rather than evaluative may help to increase people's autonomy (and decrease their resistance) when under algorithmic surveillance.

Under human surveillance, however, people might not feel as large of a difference in their autonomy when they are in evaluative (versus developmental) contexts. In developmental contexts, individuals use the information they receive from the surveillance for themselves; there are no external consequences. However, when people receive developmental feedback from a human (as opposed to AI), this introduces the possibility of negative judgment[34]. Since someone else has reviewed their performance, this may feel somewhat like an evaluation. Individuals subjected to human surveillance in developmental contexts know that even if they are not facing external consequences (as they would in an evaluative context where they will be subjected to judgment regardless of surveillance source), the person surveilling them might still form a judgment. In support of this, research by Raveendhram and Fast (2021)[35] found that people felt more autonomous under and accepting of behavior-tracking systems (e.g., a smart badge) when they use algorithmic monitoring and feedback as opposed to algorithmic monitoring and human feedback in more developmental contexts. Based on the above arguments, we predict that when surveillance is conducted in a developmental (vs. evaluative) context, the difference in autonomy and resistance between algorithmic and human surveillance will be reduced.

## Methods

We test these hypotheses across four experimental studies in which we manipulate how participants are monitored and evaluated (i.e., algorithmic vs. human surveillance) and assess how the surveillance type affects participants' perceptions of autonomy and engagement in resistance behaviors. In Study 1 ($N = 107$), we test our hypotheses in a recall study, examining whether people who recalled algorithmic (vs. human) surveillance in their lives report feeling less autonomous and engaging in more resistance. In Study 2 ($N = 157$), we test the hypothesized main effect on resistance in a controlled lab study—examining whether participants subjected to algorithmic (vs. human) surveillance are more likely to criticize the surveillance and reduce their performance. In Study 3 ($N = 117$), we replicate and extend the results of Studies 1 and 2 by testing our hypothesized mechanism, perceived autonomy, in a controlled lab study. In Study 4 ($N = 814$), we test a potential intervention and examine how to attenuate the negative effects of algorithmic (vs. human) surveillance. Specifically, we examine whether framing the monitoring by algorithmic surveillance as developmental instead of evaluative (and therefore as more informational and less controlling[17,33]) reduces participants' lessened autonomy and reduces their increased resistance when under algorithmic (vs. human) surveillance.

In all studies, the sample size was determined before data collection, and all analyses used two-tailed tests and were performed after data collection was completed. We report all variables, manipulations, measures, data exclusions, and sample size rationales. All data and materials are available on the Open Science Framework (https://osf.io/3ztpm/?view_only=91c58da2216f4633b98d5fa1cb849808), and pre-registrations can be found on aspredicted.org. We report all pre-registered analyses in the main text or the Supplementary Online Materials (SOM). Data distributions were visually inspected for normality, but this was not formally tested (see Supplemental Note 8 and Supplemental Figs. 1–7 for data visualizations and replications of results using transformations and non-parametric tests for

statistical tests that assume normality). This research was conducted in accordance with established ethical guidelines (e.g., informed consent was established verbally for Studies 2 and 3 and written for Studies 1, 3, and 4, participant data was de-identified, and all participants were debriefed) and was approved by the Institutional Review Board at Cornell University.

## Study 1: Participants
We recruited 120 participants via Cloud Research. We determined this sample size based on the effect size we obtained in a pilot study, our desire for 80% power, and our expected number of exclusions. To promote data quality, participants were required to hold a 95% approval rating, reside in the United States, and be fluent in English. To gain access to the survey, participants were also required to pass a simple CAPTCHA (Completely Automated Public Turing test to tell Computers and Humans Apart) and two pre-screening questions that asked if they had experienced both human and algorithmic surveillance. We then excluded participants who completed the survey if they failed to write about an event, described an event that was off-topic, or wrote a response that was not interpretable, leaving a final sample of 107 participants (50.5% identified as women, 47.7% as men, and 1.8% as non-binary/gender non-conforming; 11.2% as Black/African American, 6.5% as Asian/Asian American/Pacific Islander, 70.1% as White/European American, 1.9% as Latino/Hispanic American, 1.9% as Middle Eastern/Arab American, and 8.4% as Biracial/Mixed-Race; $M_{age}$ = 37.09, $SD_{age}$ = 11.06). Participants were compensated $0.75 for participating. Sample size, predictions, exclusion criteria, and analysis plans were pre-registered at As.predicted.org (#114747) on 11/30/2022.

## Study 2: Participants
We recruited 157 total participants (54.8% identified as women, and 45.2% as men; 5.1% as Black/African American, 17.2% as Asian/Asian American/Pacific Islander, 45.2% as White/European American, 7.6% as Latino/Hispanic American, 3.2% as Middle Eastern/Arab American, and 21.7% as Biracial/Mixed-Race; $M_{age}$ = 18.57, $SD_{age}$ = 1.17) from a large research university in the Northeastern United States. Since it was not possible to determine exactly how many people would choose to participate, we aimed to collect as many participants as possible within a pre-determined time period, with the goal of recruiting at least 50 participants per cell[36]. The participants received extra credit for taking part in the experiment. This study was not pre-registered.

## Study 3: Participants
We recruited 127 participants (70% identified as women, 25.6% as men, 0.9% as cisgender men, 1.8% as cisgender women, .9% as non-binary/gender non-conforming, and 0.9% did not disclose; 5.1% as Black/African American, 38.5% as Asian/Asian American/Pacific Islander, 39.3% as White/European American, 1.7% as Latino/Hispanic American, 0.9% as Middle Eastern/Arab American, 13.6% as Biracial/Mixed-Race, and 0.9% did not disclose; $M_{age}$ = 22.57, $SD_{age}$ = 8.11) from a large research university in the Northeastern United States. Since it was not possible to determine exactly how many people would choose to participate, we aimed to collect as many participants as possible within a pre-registered, pre-determined time period, with the goal of recruiting at least 50 participants per cell[36]. Due to some technology failures, we were unable to collect data from 10 participants, leaving a final sample size of 117 participants. For taking part in the experiment, participants received extra credit or a 5-dollar gift card and a chance to win an additional 25-dollar gift card. Sample size, predictions, exclusion criteria, and analysis plans were pre-registered at As.predicted.org (#65208) on 05/05/2021.

## Study 4: Participants
We recruited 814 participants (58.8% identified as women, 39.2% as men, 0.8% as non-binary/gender non-conforming, 0.2% as agender, 0.1% as cisgender women, 0.1% as cisgender men, 0.2% as transmasculine, and 0.6% did not disclose; 10.3% as Black/African American, 6.3% as Asian/Asian American/Pacific Islander, 71.9% as White/European American, 4.8% as Latino/Hispanic American, 0.2% as Middle Eastern/Arab American, .1% as Native American, 6.3% as Biracial/Mixed-Race, and 0.1% did not disclose; $M_{age}$ = 39.99, $SD_{age}$ = 12.25) via Cloud Research, based on our recruiting heuristic of 200 participants per cell. To promote data quality, participants were required to hold a 95% approval rating, reside in the United States, and be fluent in English. Participants were also required to pass a simple CAPTCHA check to gain access to the survey. Participants were compensated $0.60 for participating. Sample size, predictions, exclusion criteria, and analysis plans were pre-registered at As.predicted.org (#113436) on 11/17/2022.

## Study 1: Procedure
Study 1 served as an initial test of the hypothesized effects. Specifically, we surveyed people who had experienced monitoring and evaluation by both human and algorithmic surveillance.

We asked participants to recall and then write about a time when they were monitored and evaluated by AI technology (i.e., algorithmic surveillance) and a time when they were monitored and evaluated by a person (i.e., human surveillance) in a random order (i.e., a within-subjects design). Participants were asked to "Bring to mind a time, from your own life, where you were monitored and evaluated by [artificial intelligence technology/a person]." In both conditions, participants were asked to write about their experiences and complete the dependent variable measures. Finally, participants answered several demographic questions (see OSF for full materials and Supplementary Note 1).

## Study 2: Procedure
In Study 2, we further tested the hypothesized main effect of surveillance type on resistance in a controlled lab study. Participants completed the experiment on Zoom, a video conference software system. Participants were randomly assigned to one of two conditions—the algorithmic surveillance condition ($n$ = 81) or the human surveillance condition ($n$ = 76)—and engaged in the real-time surveillance of performance task in which they were asked to complete an idea generation task while they were surveilled in real-time.

We conducted the study in two phases. In the first phase of the study, participants in each time slot were asked to work together as a group in a breakout room to come up with as many themes for a theme park as they could. Participants were told they would be monitored and evaluated by either a research assistant or artificial intelligence (depending on the condition) that would pay attention to their interactions and responses on the document in real-time. To reinforce the manipulation, participants in the algorithmic surveillance condition saw that an "AI Technology Feed" was in their meeting while they were completing the task (this was a Zoom account we created for the purposes of this study). Conversely, participants in the human surveillance condition met a research assistant over Zoom at the beginning of the study who would be checking on them. After participants had worked on the task for exactly three minutes, they were provided with an evaluative message, which stated that they were not coming up with enough ideas and should try to come up with more ideas. In the algorithmic surveillance condition, this message was broadcasted into the Zoom breakout room by the AI Technology Feed. In the human surveillance condition, the research assistant entered the Zoom breakout room and delivered the message.

In the second phase of the study, participants were asked to complete another idea-generation task individually. Specifically, participants were instructed to choose their favorite theme park idea from the first round and to choose a segment of the theme park from a pre-set list of options (e.g., kid rides, thrill rides, etc.). After participants selected their segment, they worked individually to come up with as many attractions for their theme park segment as possible. They were given exactly five minutes to complete this task. After five minutes, participants stopped working on the task, and they were moved to a survey where they filled out other measures and a few demographic questions (see Supplementary Notes 3 and 4 and Supplementary Table 3 for information about a couple of additional measures, including perceived privacy invasion).

## Study 3: Procedure

In Study 3, we aimed to replicate the results of Study 2 while controlling for potential confounds. We also assessed whether real-time monitoring by algorithmic (vs. human) surveillance led to reduced perceived autonomy. As in Study 2, participants were put under human ($n = 57$) or algorithmic ($n = 60$) surveillance while completing the idea generation task. The surveillance procedure was almost identical to Study 2 but with the following changes: (1) in both conditions, a Zoom feed was utilized to indicate the surveillance source, with either the research assistant's name or "AI Technology Feed" displayed (but no video), (2) in both conditions, the Zoom feeds remained in the breakout room for the entire duration of the tasks to control for monitoring duration, (3) the evaluative messages in both conditions were sent as a written broadcast to reduce differences between conditions due to verbal vs. written message delivery, and (4) a measure of perceived autonomy was included. As in Study 2, we additionally measured participants' perceived privacy invasion (the pattern of results follows the pattern of results for perceived autonomy; please see Supplementary Notes 5–7 and Supplementary Table 7).

## Study 4: Procedure

The goal of Study 4 was to examine if framing surveillance as developmental instead of evaluative (and therefore as more informational and less controlling[17,33]) would increase participants' perceived autonomy (and consequently lessen their resistance). Although we predicted that the developmental framing would be beneficial overall, we predicted that it would be especially beneficial for algorithmic surveillance as it would increase participants' perceived autonomy by removing any source of human judgment, in line with previous research[35]. Therefore, in this study, we implemented a 2 (Source: algorithmic vs. human surveillance)×2 (Purpose: evaluative vs. developmental feedback) between-subjects design. Consistent with the previous studies, we predicted that participants who were surveilled in a work context for evaluative purposes by algorithmic surveillance, compared to human surveillance, would perceive that they had less autonomy and would resist more (in terms of their intention to quit). Critical to this study, we also predicted that these differences by surveillance source would be attenuated when surveillance was conducted for developmental purposes.

We asked participants to picture themselves working as telephone sales representatives in which their primary responsibilities included selling products and services to customers over the phone. Next, participants were randomly assigned to one of the following four conditions from our 2 × 2 design: the evaluative human surveillance condition, the evaluative algorithmic surveillance condition, the developmental human surveillance condition, or the developmental algorithmic surveillance condition. Participants in the *evaluative conditions* were told that a sample of their completed calls would be analyzed (by either an algorithm or a human, depending on the source condition) to evaluate their performance. Conversely, participants in the *developmental conditions* were told that a sample of their completed calls would be analyzed (again by either an algorithm or a human) to provide them with developmental feedback on their performance. Participants then completed measures of perceived autonomy, intentions to quit, and several demographic variables.

## Study 1: Measures

To measure participants' perceived autonomy, we used a 4-item scale. Our scale was modified from a pre-established scale[35] to fit the current context. Example items include, "The monitoring and evaluation by the [source] made me feel controlled" and "The monitoring and evaluation by the [source] made me feel like I was being too closely observed, monitored, and evaluated" (algorithmic surveillance $\alpha = 0.82$; human surveillance $\alpha = 0.75$). These items were rated on a 7-point Likert scale (1 = *Strongly disagree*, 7 = *Strongly agree*).

To measure participants' desire to resist the surveillance, we used a 4-item scale created for the purpose of this study that asked participants the extent to which they wanted to directly criticize the monitoring and

evaluation form, try to "game" the [system/person], avoid the monitoring and evaluation, and find a different organization that doesn't utilize monitoring and evaluation (algorithmic surveillance $\alpha = 0.92$; human surveillance $\alpha = 0.91$). These items were rated on a 7-point Likert scale (1 = *Strongly disagree*, 7 = *Strongly agree*).

## Study 2: Measures

We captured participants' engagement in resistance behaviors in two ways. First, we provided participants with an opportunity to give feedback about the study. We then coded their responses for criticism of the surveillance. Second, we measured participants' performance by the number of ideas the participants generated for their segment of the theme park (completed individually). Reductions in performance could be caused by reactance, as participants were told that the purpose of the surveillance was to ensure they were coming up with enough ideas.

## Study 3: Measures

To measure participants' perceived autonomy, we used the same pre-established scale[35] from Study 1 ($\alpha = 0.81$). As in Study 2, we assessed whether participants criticized the surveillance when given the option to provide feedback about the study. We also assessed individual performance on the idea generation task.

## Study 4: Measures

To measure participants' perceived autonomy, we used the same pre-established scale[35] from Studies 1 and 3 ($\alpha = 0.89$). To measure participants' resistance to the surveillance, we captured their intentions to quit using three items from a pre-established scale[37]. The items included, "I would actively look for a job outside of this company," "As soon as I could find a different job, I would leave this company," and "I would seriously think about quitting this job" ($\alpha = 0.95$). These items were rated on a 7-point Likert scale (1 = *Strongly disagree*, 7 = *Strongly agree*).

## Reporting summary

Further information on research design is available in the Nature Portfolio Reporting Summary linked to this article.

## Results

### Study 1: Results

As predicted, when participants recalled a time when they were put under algorithmic surveillance, they perceived they had less autonomy ($M = 2.81$, $SD = 1.39$) than when they recalled a time when they were put under human surveillance ($M = 3.32$, $SD = 1.38$), $t(106) = -3.33$, $p = 0.001$, $d = -0.32$, 95% CI [−0.52, −0.13].

Also, as predicted, when participants recalled a time when they were put under algorithmic surveillance, they reported a greater desire to engage in resistance behaviors ($M = 3.54$, $SD = 1.83$) than when they recalled a time when they were put under human surveillance ($M = 2.93$, $SD = 1.79$), $t(106) = 3.22$, $p = 0.002$, $d = 0.31$, 95% CI [0.12, 0.51].

To test the hypothesis that surveillance source (algorithmic vs. human) affects resistance behaviors through perceived autonomy, we ran a simple mediation analysis using the Montoya and Hayes (2017) bootstrapping procedure (with 10,000 iterations) for within-subjects mediation[38]. When both surveillance source (algorithmic vs. human) and perceived autonomy were entered into a linear regression model predicting the desire to engage in resistance behaviors, surveillance source (algorithmic vs. human) was no longer significant ($b = 0.30$, se[$b$] = 0.17), $t(104) = 1.73$, $p = 0.086$, whereas perceived autonomy was a significant predictor of the desire to engage in resistance behaviors ($b = -0.59$, se[$b$] = 0.10), $t(104) = -5.69$, $p < 0.001$. Further, the bootstrapping analysis revealed that the indirect effect through perceived autonomy reached significance, *indirect effect* = 0.30, $SE = 0.12$, 95% CI [0.11, 0.55]. These results are in line with our hypothesis that participants wanted to resist algorithmic surveillance more because they perceived they were less autonomous when subjected to it. Of course, mediation analyses are limited, and we cannot be confident in the causal

order of our mediator and dependent variable. However, our results were consistent with the hypothesis that lower perceived autonomy explained why participants engaged in greater resistance when put under algorithmic surveillance as opposed to human surveillance, and we believe that this order of variables has the strongest theoretical support[5,13,17,39].

Study 1 provided preliminary support for our primary hypotheses. When participants recalled their lived experience with algorithmic surveillance, they perceived they had less autonomy and subsequently reported a higher desire to engage in resistance behaviors than when they recalled their lived experience with human surveillance.

## Study 2: Results

As hypothesized, participants who were put under algorithmic surveillance criticized the surveillance more (30.9%) than participants who were put under human surveillance (6.6%), $\chi^2(1, 157) = 14.96$, $p < 0.001$, $\varphi = 0.31$, 95% bootstrapped CI [0.17, 0.44]. Examples of criticism about algorithmic surveillance include: "The reinforcement from the AI made the situation just more stressful and less creative," and "[It] wasn't that accurate, I was trying to write a lot of things." Moreover, participants in the algorithmic surveillance condition also said: "…sometimes people were just trying to think of ideas and the AI monitor went off," and "I feel like a real person monitoring the group would have been better than AI." On the other hand, examples of criticism about human surveillance include: "While the monitoring felt a little invasive, I recognize that it was necessary for the task," and "[They] kept telling us we didn't have enough ideas … it was not nice."

Further, participants who were put under algorithmic surveillance performed worse; they came up with fewer individual responses on the idea generation task ($M = 6.07$, $SD = 4.10$) than the participants who were put under human surveillance ($M = 8.37$, $SD = 4.79$), $t(155) = -3.23$, $p = 0.002$, $d = -0.52$, 95% CI [−0.83, −0.20]. Thus, in support of our hypothesis, algorithmic (vs. human) surveillance led to greater resistance, as demonstrated by increased criticism of the surveillance and worse performance. We also tested these hypotheses using mixed models and found the same pattern of results (see Supplementary Note 2 and Supplementary Tables 1 and 2 for more information).

Study 2 provided additional support for the hypothesized main effect on resistance: When put under algorithmic (vs. human) surveillance, individuals engaged in more resistance behaviors, specifically increased criticism and decreased performance. These effects were not limited to self-report measures but constituted actual behavioral measures of resistance. Thus, in a study that simulated real-world surveillance, we showed that algorithmic surveillance leads to more resistance.

Nevertheless, the results of Study 2 are subject to several limitations that we sought to address in Study 3. For example, in the human surveillance condition of Study 2, we had a human experimenter leave the Zoom breakout room and then re-enter the breakout room to deliver the evaluative message to the participants, whereas the "AI Technology Feed" was present in the Zoom room for the entirety of the task. Thus, although the participants were informed that their responses would be evaluated in real-time in both conditions (i.e., they were informed that the [experimenter or "AI Technology Feed"] had access to their responses in real-time for the entirety of the task), there were still differences in the amount of time that each surveillance source was present in the Zoom room. Moreover, the human research assistant in Study 2 delivered the evaluative message verbally, whereas the AI technology feed delivered a written message. Although these differences between algorithmic and human surveillance often play out in the real world, to increase our internal validity in Study 3, we made the two surveillance conditions as similar as possible by holding the duration of monitoring and method of message delivery constant across conditions.

## Study 3: Results

As predicted, participants who were put under algorithmic surveillance perceived they had less autonomy ($M = 3.91$, $SD = 1.03$) than participants who were put under human surveillance ($M = 4.61$, $SD = 0.99$), $t(115) = -3.78$, $p < 0.001$, $d = -0.70$, 95% CI [−1.07, −0.32].

Also, as predicted and replicating Study 2, participants who were put under algorithmic surveillance criticized the surveillance more (11.7%) than participants who were put under human surveillance (0.0%), $\chi^2(1, 117) = 7.07$, $p = 0.008$, $\varphi = 0.25$, 95% bootstrapped CI [0.14, 0.34]. Further, as predicted and replicating Study 2, participants who were put under algorithmic surveillance came up with fewer individual responses on the idea generation task ($M = 6.18$, $SD = 2.98$) than the participants who were put under human surveillance ($M = 7.74$, $SD = 4.16$), $t(115) = -2.33$, $p = 0.021$, $d = -0.43$, 95% CI [−0.80, −0.06].

We ran two simple mediation analyses using the Hayes (2018) bootstrapping procedure (with 10,000 iterations) to test our hypothesis that surveillance source (algorithmic vs. human) affects resistance behaviors—criticism (model 1) and performance (model 2)—through perceived autonomy[40]. For model 1, when both surveillance source (algorithmic vs. human) and perceived autonomy were entered into a logistic regression model predicting criticism of the surveillance, surveillance source (algorithmic vs. human) was no longer significant ($b = 14.38$, se[$b$] = 666.42), $Z = 0.02$, $p = 0.983$, whereas perceived autonomy was a significant predictor of criticism ($b = -1.16$, se[$b$] = 0.50), $Z = 2.30$, $p = 0.022$. Further, a bootstrapping analysis revealed that, for model 1, the indirect effect through perceived autonomy reached significance, *indirect effect* = 0.82, $SE = 6.60$, 95% CI [0.08, 3.06].

For model 2, when both surveillance source (algorithmic vs. human) and perceived autonomy were entered into a linear regression model predicting performance, surveillance source (algorithmic vs. human) remained significant ($b = -1.51$, se[$b$] = 0.71), $t(114) = -2.13$, $p = 0.035$, whereas we found no statistically significant evidence that perceived autonomy predicted performance ($b = 0.06$, se[$b$] = 0.33), $t(114) = 0.18$, $p = 0.858$. Further, a bootstrapping analysis revealed that, for model 2, the indirect effect through perceived autonomy failed to reach significance, *indirect effect* = −0.04, $SE = 0.21$, 95% CI [−0.41, 0.43].

Study 3 replicated and extended the results of Study 2 while controlling for potential confounds by testing and finding evidence in line with the hypotheses that real-time algorithmic (vs. human) surveillance would decrease autonomy and increase resistance behaviors. Even when the duration of monitoring time was held constant, and both conditions utilized alternative Zoom feeds to deliver messages in the same way, participants engaged in more resistance behaviors (increased criticism and reduced performance) and perceived they had less autonomy when put under algorithmic (vs. human) surveillance.

We further found evidence partially in line with the hypothesized mechanism in that reduced autonomy seemed to account for the increased criticism in response to algorithmic (vs. human) surveillance but not worse performance, a point we return to in the general discussion.

## Study 4: Results

Table 1 shows the descriptive statistics by condition for each dependent variable. Results of a $2 \times 2$ factorial ANOVA on perceived autonomy revealed a non-significant main effect of source, $F(1, 810) = -1.39$, $p = 0.239$, $d = -0.17$, 95% CI [−0.44, 0.11]. There was a significant main effect of purpose, $F(1, 810) = -25.46$, $p < 0.001$, $d = -0.71$, 95% CI [−0.99, −0.43], such that the developmental purpose led to higher autonomy overall, and a significant interaction, $F(1, 810) = -9.16$, $p = 0.003$, $d = -0.43$, 95% CI [−0.70, −0.15]. Replicating the results from Studies 1 and 3, when participants were put under surveillance for evaluative reasons, participants in the algorithmic surveillance condition perceived they had less autonomy than participants in the human surveillance condition, $t(810) = -3.03$, $p = 0.003$, $d = -0.30$, 95% CI [−0.49, −0.10]. However, when participants were put under surveillance for developmental reasons, participants no longer reported less perceived autonomy in the algorithmic surveillance condition, $t(810) = 1.28$, $p = 0.200$, $d = 0.13$, 95% CI [−0.07, 0.33].

Results of a $2 \times 2$ factorial ANOVA on intention to quit revealed a non-significant main effect of source, $F(1, 810) = 0.11$, $p = 0.746$, $d = 0.05$, 95% CI [−0.23, 0.32], a significant main effect of purpose, $F(1, 810) = 29.09$,

**Table 1 | Means and standard deviations (in parentheses) per condition in Study 4**

| Dependent variable | Evaluative | | Developmental | |
|---|---|---|---|---|
| | Algorithmic surveillance (*n* = 220) | Human surveillance (*n* = 202) | Algorithmic surveillance (*n* = 194) | Human surveillance (*n* = 198) |
| Perceived autonomy | 2.49 (1.10) | 2.85 (1.17) | 3.18 (1.32) | 3.03 (1.31) |
| Intention to quit | 5.20 (1.54) | 4.92 (1.56) | 4.34 (1.70) | 4.55 (1.67) |

**Table 2 | Coefficients from moderated mediation analysis in Study 4**

| | ME (perceived autonomy) | | | Y (intentions to quit) | | |
|---|---|---|---|---|---|---|
| | *b* [95% CI] | *t* | *p* | *b* [95% CI] | *t* | *p* |
| Constant | 2.89 [2.80, 2.97] | 67.12 | 0.000 | 7.39 [7.19 7.60] | 70.38 | 0.000 |
| Surveillance source (algorithmic vs. human) | −0.05 [−0.14, 0.03] | −1.18 | 0.239 | −0.03 [−0.11, 0.05] | −0.68 | 0.497 |
| Purpose (evaluate vs. develop) | −0.22 [−0.30, −0.13] | −5.05 | 0.000 | 0.11 [0.03, 0.19] | 2.59 | 0.010 |
| Surveillance source × purpose | −0.13 [−0.21, −0.05] | −3.03 | 0.003 | 0.00 [−0.08, 0.09] | 0.11 | 0.910 |
| Perceived autonomy | | | | −0.91 [−0.98, −0.85] | −27.27 | 0.000 |

$p < 0.001$, $d = 0.76$, 95% CI [0.48, 1.03], such that the developmental purpose led to lower intentions to quit overall, and a significant interaction, $F(1, 810) = 4.74$, $p = 0.030$, $d = 0.31$, 95% CI [0.03, 0.58]. When participants were put under surveillance for evaluative reasons, participants in the algorithmic surveillance condition reported a marginally greater intention to quit (thus resisting the surveillance more) than participants in the human surveillance condition, $t(810) = 1.80$, $p = 0.072$, $d = 0.18$, 95% CI [−0.02, 0.37]. However, when participants were put under surveillance for developmental reasons, participants no longer reported greater intentions to quit in the algorithmic surveillance condition, $t(810) = −1.29$, $p = 0.199$, $d = −0.13$, 95% CI [−0.33, 0.07].

To examine if perceived autonomy mediated the interactive effect between surveillance source (algorithmic vs. human) and purpose (evaluative vs. developmental) on intentions to quit, we conducted a moderated mediation analysis using Hayes's (2018)[40] PROCESS macro (Model 8), with 10,000 bootstrapping resamples. We entered surveillance source (algorithmic = 1, human = −1) as our independent variable, purpose (evaluation = 1, development = −1) as our moderator, perceived autonomy as our mediator, and intentions to quit as our dependent variable. Figure 1 and Table 2 show the results for the two multiple regressions.

As predicted, this analysis revealed a significant indirect effect of the interaction of surveillance source and purpose on intentions to quit though perceived autonomy: *Index of Moderated Mediation* = 0.24, *SE* = 0.08, 95% CI [0.08, 0.39]. When the surveillance was implemented to evaluate, there was a significant indirect effect of surveillance source (AI vs. human) on intentions to quit through perceived autonomy, *indirect effect* = 0.17, *SE* = 0.05, 95% CI [0.06, 0.27]. When the surveillance was implemented to provide developmental feedback, however, the indirect effect through perceived autonomy did not reach significance, *indirect effect* = −0.07, *SE* = 0.06, 95% CI [−0.19, 0.05]. Thus, the mediation results are consistent with the idea that when monitoring is used for evaluation, people intend to quit their jobs more when the monitoring source is algorithmic (vs. human) because they perceive they have less autonomy. However, when monitoring is conducted solely for personal development, people do not intend to quit more when the monitoring source is algorithmic (vs. human) because, in developmental contexts, they do not perceive they have significantly less autonomy.

Taken together, in Study 4, we again replicated the results of Studies 1–3, showing that when people are subjected to algorithmic (vs. human) surveillance in an evaluative context, they report lower perceived autonomy and a higher intention to resist (significant and marginal, respectively, in this study). It is interesting to note that even though real-time surveillance did not occur, and the same amount of information was collected in both

conditions (i.e., a sample of pre-recorded calls), the perceived differences between human and algorithmic evaluative judgment still seemingly affected how autonomous participants anticipated they would feel.

The results of Study 4 also supported the hypothesis that framing monitoring by algorithmic surveillance as developmental instead of evaluative helps to alleviate the negative effects that individuals experience when subjected to algorithmic (vs. human) surveillance. Specifically, when the surveillance was framed as developmental, participants no longer reported statistically significant differences in perceived autonomy as a result of algorithmic (vs. human) surveillance, and they also did not report statistically significant differences in terms of intentions to quit (our measure of resistance in this study). In fact, the direction of effects was trending in the opposite direction, such that in the developmental conditions, participants reported greater perceptions of autonomy and lower intentions to quit in the algorithmic surveillance condition as compared to the human surveillance condition. Therefore, it seems that if people think that AI will only be used to provide them with developmental feedback to help them in their self-determined endeavors and will not be used to evaluate them, then they do not feel as limited and controlled by it and do not resist it to the same extent.

## Discussion

In this paper, we explored how using algorithmic (vs. human) surveillance to monitor and evaluate individuals impacts their perceptions of autonomy and engagement in resistance. In support of our hypotheses, we found that participants who were put under algorithmic (vs. human) surveillance perceived that they had less autonomy (Studies 1, 3, and 4) and engaged in greater resistance (possibly to restore their autonomy) by complaining more, performing worse, and intending to quit (Studies 1–4). Furthermore, we found that framing the algorithmic surveillance as developmental rather than evaluative increased participants' perceived autonomy and decreased their intentions to resist (Study 4). Thus, while people often react more negatively to algorithmic (vs. human) surveillance, there can be times when people are more accepting of it.

Our paper makes several contributions. From a practical standpoint, we show that intentions to implement and use algorithmic surveillance to monitor and evaluate individuals can result in unintended consequences such as reduced perceived autonomy and increased resistance. Although teachers, managers, and law enforcement may want to implement algorithmic surveillance to get people to behave in the way they want, they likely also want to avoid the resistance behaviors we demonstrated (e.g., reduced performance, increased criticism, and increased intentions to quit). However, these negative consequences are not guaranteed to result from

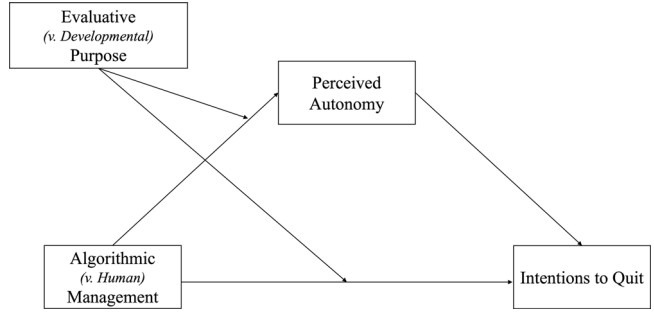

**Fig. 1 | Visualization of hypothesized moderated mediation in Study 4.** This figure portrays the hypothesized moderated mediation tested in Study 4, examining the interactive effect between surveillance source (algorithmic vs. human) and purpose (evaluative vs. developmental) on intentions to quit through perceived autonomy.

implementing algorithmic surveillance. We demonstrate a potential way algorithmic surveillance can be utilized that attenuates some of the unintended consequences that can arise—specifically, a developmental framing —that decision-makers can consider when implementing algorithmic surveillance.

Theoretically, our research contributes to the understanding of the psychology of technology, adding to the growing body of work on perceptions of AI and algorithmic decision-making[18,22,24,35,41–46]. Although people often think algorithmic judgment is less biased than human judgment[45,47], we found that people resist algorithmic judgment when it is used to monitor and evaluate them. Our results are likely due to people's tendency to overestimate the extent to which human decision-makers understand their internal states, such as the intentions guiding their behaviors[25,26], and their belief that algorithmic surveillance lacks the ability to take context[19] or uniqueness[21] into account. Thus, when individuals are subjected to algorithmic (vs. human) surveillance, they likely perceive that they have limited choice in how to behave (otherwise, the algorithm will misinterpret their behavior), which diminishes their perceived autonomy, leading them to resist algorithmic (vs. human) surveillance to a greater extent.

## Limitations
We found consistent evidence that algorithmic (vs. human) surveillance led participants to perceive they had less autonomy and engage in more resistance behaviors. However, our research is subject to a few limitations that future work should address. First, while the nature of our experimental designs enabled us to test the causal effects of surveillance sources in controlled settings, future research should be conducted within more realistic organizational contexts to strengthen the ecological validity of the present findings. Although we were able to assess real-time surveillance in a controlled lab setting, future research should assess real-time surveillance in an organizational setting where there are meaningful rewards and punishments for behavior. Future research in an organizational setting would also be better positioned to capture a greater range of resistance behaviors, along with the long-term effects of algorithmic (vs. human) surveillance.

Theoretically and empirically, it seems that participants' reduced sense of autonomy as a result of algorithmic surveillance leads to their increased engagement in resistance behaviors. However, mediation analyses are limited, and it is possible that reduced perceived autonomy is just one of several mechanisms at play. In fact, in Study 3, performance was not accounted for by perceived autonomy. Thus, perhaps participants subjected to algorithmic surveillance do not perform worse to restore control, as we had expected, but instead for another reason. People might resist algorithmic surveillance because they feel uneasy about it due to its coldness[47–50] or because they have trouble concentrating when subjected to it[51]. Future work ought to consider these alternative explanations for the observed effect. Relatedly, future research should examine how the present findings relate to

other theories on autonomy and agency, such as discussions around agency in embodied cognitive science[52,53] as well as the wider philosophical debates regarding surveillance and its impact on individual agency[6,7,54].

Additionally, many individual differences predict greater or lesser sensitivity to perceived reductions in autonomy[32,55] and willingness to comply with various policies[56]. Individual differences such as psychological entitlement, agreeableness, and trait reactance, among others, may predict who is more likely to perceive that their autonomy is threatened by algorithmic surveillance and show a lesser willingness to comply with algorithmic surveillance in general.

Another important limitation of this work relates to constraints on generality[57]. It is important to note that across all samples, the participants could be classified as "WEIRD" (Western, Educated, Industrialized, Rich, and Democratic)[58]. Thus, the present findings may not fully generalize to samples with different demographic compositions. Future research should evaluate the current findings in a variety of different samples, especially in more diverse and representative samples.

Moreover, considering the ever-evolving landscape of technology, it is imperative for future research to continually replicate, reassess, and analyze the influence of future technological developments on individuals' perceptions of autonomy and engagement in resistance to algorithmic (vs. human) surveillance. As technology continues to evolve, the nature and capabilities of algorithmic surveillance are likely to become more sophisticated, which may lead to a shift in how individuals perceive and react to algorithmic surveillance. Relatedly, as the public discourse surrounding these technological developments continues to evolve, individuals' perceptions of and reactions to algorithmic (vs. human) surveillance may also undergo significant changes (e.g., the studies conducted in this manuscript were conducted before the release of publicly accessible Large Language Models such as ChatGPT). Future research should consider how these dynamic factors—technological developments and evolving societal discourse—might interplay and reshape how individuals perceive and react to algorithmic surveillance.

Finally, considering that the use of algorithmic surveillance is becoming common throughout society, and perceptions of autonomy are essential for individual well-being[9,59], investigating additional ways to attenuate individuals' perceptions of reduced autonomy would have profound theoretical and practical significance. Although we found one way to attenuate individuals' perceptions of reduced autonomy from algorithmic (vs. human) surveillance, specifically a developmental framing, we recognize that a developmental framing will not work in all contexts. For example, it would be hard to frame facial recognition software aimed at detecting unethical behavior in a developmental way. Further, if surveillance is framed as developmental but used in evaluative ways, this may lead to extensive backlash. Therefore, future research should investigate other ways to attenuate individuals' perceptions of reduced autonomy from algorithmic surveillance in these contexts.

## Conclusions
In society, people are routinely subjected to surveillance to control or influence their behaviors[60,61]. Given that the use of algorithmic surveillance is increasingly replacing prototypical human surveillance[1–3], it is ever more important to study the unintended costs that could severely impact individuals and society. Our paper offers initial insight—finding that individuals monitored and evaluated by algorithmic (vs. human) surveillance perceive that their autonomy is reduced to a greater extent, report greater intentions to resist the surveillance, criticize the surveillance more, and perform worse. However, the lack of perceived autonomy and increased resistance can be alleviated if algorithmic surveillance is viewed as developmental. Thus, algorithmic surveillance will not necessarily lead to adverse outcomes for individuals—it just depends on how it is implemented.

## Data availability
All data is available for download and reanalysis on https://osf.io/3ztpm/?view_only=91c58da2216f4633b98d5fa1cb849808.

## Code availability

All codes for reproducing the analyses and study materials for reproducing the experiments are located at https://osf.io/3ztpm/?view_only=91c58da2216f4633b98d5fa1cb849808. All analyses were conducted on SPSS version 29 or R version 4.0.2.

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

## Acknowledgements
The authors would like to thank Michelle Cao, Emily Mason, Joshua Samuel, Jemima Yoon, the members of the Expo Lab at Cornell ILR, and the Logg Lab at Georgetown McDonald School of Business for their valuable time and feedback on this work. The authors received no specific funding for this work.

## Author contributions
Rachel Schlund: Conceptualization, methodology, investigation, formal analysis, project administration, writing—original draft. Emily M. Zitek: Conceptualization, methodology, investigation, validation, writing—review and editing.

## Competing interests
The authors declare no competing interests.
