## [Peer Review File · Communications Psychology]

2nd Nov 23

Dear Ms Schlund,

Thank you for your patience during the peer-review process. Your manuscript titled "I know I'm Being Watched: Algorithmic (v. Human) Surveillance Leads to Lower Perceptions of Autonomy and Increased Resistance" has now been seen by 2 reviewers, and I include their comments at the end of this message. They find your work of interest, but raised some important points. We are interested in the possibility of publishing your study in *Communications Psychology*, but would like to consider your responses to these concerns and assess a revised manuscript before we make a final decision on publication.

We therefore invite you to revise and resubmit your manuscript, along with a point-by-point response to the reviewers. Please highlight all changes in the manuscript text file.

Editorially, we consider it important that the revised manuscript addresses the requests for consideration of the ethical and ecological validity concerns raised by Reviewer 1, as well as an acknowledgment of the methodological limitation raised by Reviewer 2.

Please note that your revised manuscript must comply with our formatting and reporting requirements, which are summarized on the following checklist:

Communications Psychology formatting checklist and also in our style and formatting guide Communications Psychology formatting guide.

Please use the following link to submit your revised manuscript, point-by-point response to the referees' comments (which should be in a separate document to any cover letter) and the completed checklist:

[link redacted]

Please do not hesitate to contact me if you have any questions or would like to discuss these

revisions further. We look forward to seeing the revised manuscript and thank you for the opportunity to review your work.

Best regards,

Fernando Marmolejo-Ramos

Fernando Marmolejo-Ramos, PhD
Editorial Board Member
Communications Psychology
orcid.org/0000-0003-4680-1287

EDITORIAL POLICIES AND FORMATTING

Editorial Policy: Policy requirements (Download the link to your computer as a PDF.)

* **CODE AVAILABILITY:** All Communications Psychology manuscripts must include a section titled "Code Availability" at the end of the methods section. In the event of publication, we require that the custom analysis code supporting your conclusions is made available in a publicly accessible repository; at publication, we ask you to choose a repository that provides a DOI for the code; the link to the repository and the DOI will need to be included in the Code Availability statement. Publication as Supplementary Information will not suffice. We ask you to prepare code at this stage, to avoid delays later on in the process.

* **DATA AVAILABILITY:**

All Communications Psychology manuscripts must include a section titled "Data Availability" at the end of the Methods section or main text (if no Methods). More information on this policy, is available at <http://www.nature.com/authors/policies/data/data-availability-statements-data-citations.pdf>.

At a minimum the Data availability statement must explain how the data can be obtained and whether there are any restrictions on data sharing. Communications Psychology strongly endorses open sharing of data. If you do make your data openly available, please include in the statement:
- Unique identifiers (such as DOIs and hyperlinks for datasets in public repositories)

- Accession codes where appropriate
- If applicable, a statement regarding data available with restrictions
- If a dataset has a Digital Object Identifier (DOI) as its unique identifier, we strongly encourage including this in the Reference list and citing the dataset in the Data Availability Statement.

We recommend submitting the data to discipline-specific, community-recognized repositories, where possible and a list of recommended repositories is provided at <http://www.nature.com/sdata/policies/repositories>.

If a community resource is unavailable, data can be submitted to generalist repositories such as figshare or Dryad Digital Repository. Please provide a unique identifier for the data (for example a DOI or a permanent URL) in the data availability statement, if possible. If the repository does not provide identifiers, we encourage authors to supply the search terms that will return the data. For data that have been obtained from publicly available sources, please provide a URL and the specific data product name in the data availability statement. Data with a DOI should be further cited in the methods reference section.

REVIEWERS' EXPERTISE:

Reviewer #1: AI alignment, AI in governance, philosophy of AI.

Reviewer #2: AI and human interactions, algorithms, data, and AI

REVIEWERS' COMMENTS:

Reviewer #1 (Remarks to the Author):

The study addresses a relevant and timely topic within the field of psychology, specifically focusing on the nuanced reactions of individuals to different forms of surveillance.

The inclusion of four experiments with a substantial sample size (N=1,208) demonstrates a robust methodology, which is essential for high-quality research. Additionally, the differentiation between human and algorithmic surveillance, as well as the innovative framing of the purpose of algorithmic surveillance, showcases a thoughtful and creative approach to the research question.

Furthermore, the findings, which indicate significant differences in participant reactions to different forms of surveillance, suggest that the study contributes valuable insights to the field. The paper's potential to inform both theoretical perspectives and practical applications in the domain of surveillance psychology further underscores its quality.

1. Replication Crisis Concerns: The paper does not explicitly address replication. It would be beneficial for the study to mention plans for replication or acknowledge the importance of replication in ensuring the robustness of the findings.

2. WEIRD Samples: The paper does not specify the demographic characteristics of the participants. If the sample is WEIRD, it would be important to acknowledge that the findings may not fully generalize to non-WEIRD populations.

3. Philosophical Debates on Agency: The paper does not delve into philosophical debates on agency in embodied cognitive science. Acknowledging the philosophical implications of the study's findings could add depth to the discussion.

Enriched Understanding of Embodied Experience: Embodied cognitive science emphasizes the inseparable connection between the mind, body, and environment. When applied to the loss of agency, it allows for a more nuanced understanding of how disruptions in bodily functions or interactions with the environment can impact one's sense of agency.

Barandiaran, X. E. (2017). Autonomy and enactivism: Towards a theory of sensorimotor autonomous agency. *Topoi*, 36, 409-430.

Popova, Y. B., & Rączaszek-Leonardi, J. (2020). Enactivism and ecological psychology: The role of bodily experience in agency. *Frontiers in Psychology*, 11, 539841.

Maiese, M. (2022). Mindshaping, enactivism, and ideological oppression. *Topoi*, 41(2), 341-354.

Contextualizing Psychological Disorders: Embodied cognitive science provides a framework for contextualizing psychological disorders that may involve a perceived loss of agency, such as certain types of schizophrenia or depersonalization disorders. It considers how disruptions in embodiment contribute to these experiences.

Maiese, M. (2022). *Autonomy, Enactivism, and Mental Disorder: A Philosophical Account*. Taylor & Francis.

Technological Interventions: In cases where technology is used to restore or enhance agency (e.g., neuroprosthetics, brain-computer interfaces), embodied cognition offers insights into how these interventions interface with the body and impact the individual's sense of agency.

Quirit, I. (2021). Material Engagement Theory and Extensive Enactivism Within the 4E Cognitive Debate: A Phenomenological Approach to Material Agency and Application to Current Technology.

Cheville, A., & Heywood, J. (2022, August). Victims of Outcomes: Towards an Enactivist Model of Technological Literacy. In 2022 ASEE Annual Conference & Exposition.

4. Concept of surveillance:

The authors do refer to some anecdotes to motivate the topic of the paper on surveillance:

“Some anecdotes suggest that people may not react positively to algorithmic surveillance. For example, in educational settings, students seem averse to the use of algorithmic surveillance as compared to traditional surveillance by their teachers.”

The authors present an intriguing exploration of the reactions of individuals to algorithmic surveillance, particularly in educational settings. While the manuscript briefly touches upon the

aversion of students to algorithmic surveillance compared to traditional methods, it would significantly benefit from a more comprehensive engagement with the extensive philosophical literature on surveillance and its connection to agency.

Specifically, the authors should consider delving deeper into the philosophical discourse surrounding surveillance and its implications for human agency. This would serve to bolster the argument that algorithmic surveillance can potentially impede individual autonomy and hinder the exercise of agency. By drawing on relevant philosophical frameworks and debates, the manuscript could provide a more robust foundation for its claims regarding the potential adverse reactions to algorithmic surveillance.

Below are only a few points in the literature:

Foucauldian Discourse:

Michel Foucault's works, particularly "Discipline and Punish" and "The Birth of Biopolitics," are foundational in understanding the relationship between surveillance, power, and agency. He explores how surveillance techniques are employed to regulate and control individuals within society.

Panoptic Surveillance:

Jeremy Bentham's concept of the Panopticon, later expanded upon by Foucault, examines the architectural design of a circular prison with a central watchtower, allowing for continuous, unidirectional surveillance. This concept has been widely discussed in relation to modern forms of surveillance.

Privacy and Autonomy:

Philosophers like Alan Westin ("Privacy and Freedom") and Judith Jarvis Thomson ("The Right to Privacy") have examined the importance of privacy in relation to individual autonomy and agency. They discuss how surveillance practices may infringe upon personal freedoms.

Sousveillance and Resistance:

Steve Mann's concept of "sousveillance" (inverse surveillance) explores how individuals can use technology to monitor authority figures or institutions. This perspective emphasizes the potential for individuals to resist and subvert traditional forms of surveillance.

Post-Structuralist Critiques:

Gilles Deleuze's and Félix Guattari's works, such as "A Thousand Plateaus," offer post-structuralist critiques of surveillance. They emphasize the rhizomatic nature of power and resistance, challenging traditional notions of agency.

Ethical Implications:

Anita L. Allen's "Unpopular Privacy: What Must We Hide?" and Helen Nissenbaum's "Privacy in Context" delve into the ethical dimensions of privacy and surveillance, considering how these practices impact individuals' moral agency.

Technological Determinism:

Albert Borgmann's "Technology and the Character of Contemporary Life" examines how technological systems, including surveillance technologies, shape human experiences and agency within modern society.

Surveillance Capitalism:

Shoshana Zuboff's "The Age of Surveillance Capitalism" provides a comprehensive analysis of how modern digital technologies and data collection practices have transformed the dynamics of surveillance, impacting individual agency in unprecedented ways.

Links with agency:

In the context of surveillance, understanding embodied cognitive science is crucial. It sheds light on how surveillance practices, whether human or algorithmic, interact with an individual's embodied experience. This perspective allows us to comprehend how being monitored may affect not only cognitive processes but also the entire embodied being, including physiological and sensory responses. It helps in recognizing that surveillance is not a purely cognitive phenomenon, but one that is deeply entwined with the physical and environmental aspects of an individual's experience. This insight can inform discussions on the ethical implications, psychological impacts, and potential interventions related to surveillance practices.

5. Ethical Considerations: It would be crucial to highlight any steps taken to ensure participant well-being, privacy, and informed consent, especially in the context of a study involving surveillance.

6. Ecological Validity: Mentioning how the study's controlled settings align with or deviate from real-world surveillance scenarios would be valuable.

7. Long-term Effects of Surveillance: Considering the lasting impact of surveillance could provide a more comprehensive understanding of its psychological consequences.

8. Alternative Explanations: Acknowledging potential alternative factors that could influence participant responses would strengthen the study's validity.

9. Technological Advancements: Considering the evolving nature of technology, it would be important to address how the study's findings might be influenced by future technological developments.

Reviewer #2 (Remarks to the Author):

The paper considers an important and timely research question, which is the impact of algorithmic surveillance on human behavior. Through four different studies, the authors provide convincing evidence this increasingly prevalent form of surveillance has a significant effect that should not be ignored in the age of AI. It is interesting to see that that the effect is negated when framing the purpose of the surveillance as informational (as opposed to evaluative), which has practical implications. The paper is well written and easy to follow, and the literature is well covered.

When I read the details of Study 2, I thought of a number of limitations due to differences that could have been avoided between the control and treatment conditions. However, I was pleased to see that the authors concluded their description of Study 2 by listing a number of limitations, which almost exactly matched my concerns. Later on, the authors presented Study 3, which adequately addressed all of my concerns regarding Study 2.

Minor comment:

In Study 1, I expected participants to be asked whether they recall an incident of being monitored (either by humans or by algorithms) before being asked about how they felt. This is because there could be many people who cannot even recall a single incident where they were monitored and evaluated by “artificial intelligence technology”. For such people, the task probably becomes more like a hypothetical question, e.g., if you were to imagine a situation where you are being monitored by an algorithm, how would you feel? Rather than: in a previous incident where you actually were monitored by an algorithm, how did you feel? Still, I don’t consider this to be a major comment, since the paper as a whole manages to provide sufficient evidence to support their claim.

Overall, I recommend the paper for publication.

Dear Members of the Review Team,

We greatly appreciate your thoughtful feedback. We believe we have addressed the issues raised. For your convenience, we have reproduced your comments in bold and have responded to each point below the reproduced text.

We greatly appreciate the time and effort you have put into reviewing our work. We believe the manuscript is improved as a result. Thank you.

Response to Reviewer 1

The study addresses a relevant and timely topic within the field of psychology, specifically focusing on the nuanced reactions of individuals to different forms of surveillance.

The inclusion of four experiments with a substantial sample size (N=1,208) demonstrates a robust methodology, which is essential for high-quality research. Additionally, the differentiation between human and algorithmic surveillance, as well as the innovative framing of the purpose of algorithmic surveillance, showcases a thoughtful and creative approach to the research question.

Furthermore, the findings, which indicate significant differences in participant reactions to different forms of surveillance, suggest that the study contributes valuable insights to the field. The paper's potential to inform both theoretical perspectives and practical applications in the domain of surveillance psychology further underscores its quality.

Thank you for your kind words. We found your comments to be very constructive, and they have very much helped us improve the paper.

(1) Replication Crisis Concerns: The paper does not explicitly address replication. It would be beneficial for the study to mention plans for replication or acknowledge the importance of replication in ensuring the robustness of the findings.

We very much believe that replication is an important part of the scientific process. We have added additional language to address the importance of replication throughout the general discussion (please see pgs. 29-31) and ensure that we explicitly encourage future work to “continually replicate the present findings to ensure robustness” (p. 30).

(2) WEIRD Samples: The paper does not specify the demographic characteristics of the participants. If the sample is WEIRD, it would be important to acknowledge that the findings may not fully generalize to non-WEIRD populations.

We agree that it is important to fully specify the demographic characteristics of the participants and to acknowledge that the findings may not fully generalize to non-WEIRD populations. Please find updated information regarding the demographic characteristics of the participants

under the sections titled “Participants” for each study. We have now included more details about the demographic characteristics of the participants. We have also added a “Constraints on Generality” (Simons et al., 2017) statement in the general discussion where we emphasize the importance of evaluating the current findings in a variety of different samples. Specifically, we have included the following:

Another important limitation of this work relates to constraints on generality (Simons et al., 2017). It is important to note that across all samples, the participants could be classified as “WEIRD” (Western, Educated, Industrialized, Rich, and Democratic; Henrich et al., 2010). Thus, the present findings may not fully generalize to non-WEIRD populations or samples with different demographic compositions. Future research should evaluate the current findings in a variety of different samples, especially non-WEIRD populations, and should continually replicate the present findings to ensure robustness. (p. 30).

(3) Philosophical Debates on Agency: The paper does not delve into philosophical debates on agency in embodied cognitive science. Acknowledging the philosophical implications of the study's findings could add depth to the discussion.

Enriched Understanding of Embodied Experience: Embodied cognitive science emphasizes the inseparable connection between the mind, body, and environment. When applied to the loss of agency, it allows for a more nuanced understanding of how disruptions in bodily functions or interactions with the environment can impact one's sense of agency.

Barandiaran, X. E. (2017). Autonomy and enactivism: Towards a theory of sensorimotor autonomous agency. *Topoi*, 36, 409-430.

Popova, Y. B., & Rączaszek-Leonardi, J. (2020). Enactivism and ecological psychology: The role of bodily experience in agency. *Frontiers in Psychology*, 11, 539841.

Maiese, M. (2022). Mindshaping, enactivism, and ideological oppression. *Topoi*, 41(2), 341-354.

Contextualizing Psychological Disorders: Embodied cognitive science provides a framework for contextualizing psychological disorders that may involve a perceived loss of agency, such as certain types of schizophrenia or depersonalization disorders. It considers how disruptions in embodiment contribute to these experiences.

Maiese, M. (2022). *Autonomy, Enactivism, and Mental Disorder: A Philosophical Account*. Taylor & Francis.

Technological Interventions: In cases where technology is used to restore or enhance agency (e.g., neuroprosthetics, brain-computer interfaces), embodied cognition offers insights into how these interventions interface with the body and impact the individual's sense of agency.

Quirit, I. (2021). Material Engagement Theory and Extensive Enactivism Within the 4E Cognitive Debate: A Phenomenological Approach to Material Agency and Application to Current Technology.

Cheville, A., & Heywood, J. (2022, August). Victims of Outcomes: Towards an Enactivist Model of Technological Literacy. In 2022 ASEE Annual Conference & Exposition.

We agree it is an important and fruitful avenue for future research to delve into philosophical debates on agency in embodied cognitive science. Largely due to space constraints, we do not dive into these debates in depth in the current manuscript, but we now encourage future research to do so in the general discussion (please see p. 30).

(4) Concept of surveillance:

The authors do refer to some anecdotes to motivate the topic of the paper on surveillance:

“Some anecdotes suggest that people may not react positively to algorithmic surveillance. For example, in educational settings, students seem averse to the use of algorithmic surveillance as compared to traditional surveillance by their teachers.”

The authors present an intriguing exploration of the reactions of individuals to algorithmic surveillance, particularly in educational settings. While the manuscript briefly touches upon the aversion of students to algorithmic surveillance compared to traditional methods, it would significantly benefit from a more comprehensive engagement with the extensive philosophical literature on surveillance and its connection to agency.

Specifically, the authors should consider delving deeper into the philosophical discourse surrounding surveillance and its implications for human agency. This would serve to bolster the argument that algorithmic surveillance can potentially impede individual autonomy and hinder the exercise of agency. By drawing on relevant philosophical frameworks and debates, the manuscript could provide a more robust foundation for its claims regarding the potential adverse reactions to algorithmic surveillance.

Below are only a few points in the literature:

Foucauldian Discourse:

Michel Foucault's works, particularly "Discipline and Punish" and "The Birth of Biopolitics," are foundational in understanding the relationship between surveillance, power, and agency. He explores how surveillance techniques are employed to regulate and control individuals within society.

Panoptic Surveillance:

Jeremy Bentham's concept of the Panopticon, later expanded upon by Foucault, examines the architectural design of a circular prison with a central watchtower, allowing for continuous, unidirectional surveillance. This concept has been widely discussed in relation to modern forms of surveillance.

Privacy and Autonomy:

Philosophers like Alan Westin ("Privacy and Freedom") and Judith Jarvis Thomson ("The Right to Privacy") have examined the importance of privacy in relation to individual autonomy and agency. They discuss how surveillance practices may infringe upon personal freedoms.

Sousveillance and Resistance:

Steve Mann's concept of "sousveillance" (inverse surveillance) explores how individuals can use technology to monitor authority figures or institutions. This perspective emphasizes the potential for individuals to resist and subvert traditional forms of surveillance.

Post-Structuralist Critiques:

Gilles Deleuze's and Félix Guattari's works, such as "A Thousand Plateaus," offer post-structuralist critiques of surveillance. They emphasize the rhizomatic nature of power and resistance, challenging traditional notions of agency.

Ethical Implications:

Anita L. Allen's "Unpopular Privacy: What Must We Hide?" and Helen Nissenbaum's "Privacy in Context" delve into the ethical dimensions of privacy and surveillance, considering how these practices impact individuals' moral agency.

Technological Determinism:

Albert Borgmann's "Technology and the Character of Contemporary Life" examines how technological systems, including surveillance technologies, shape human experiences and agency within modern society.

Surveillance Capitalism:

Shoshana Zuboff's "The Age of Surveillance Capitalism" provides a comprehensive analysis of how modern digital technologies and data collection practices have transformed the dynamics of surveillance, impacting individual agency in unprecedented ways.

Links with agency:

In the context of surveillance, understanding embodied cognitive science is crucial. It sheds light on how surveillance practices, whether human or algorithmic, interact with an individual's embodied experience. This perspective allows us to comprehend how being monitored may affect not only cognitive processes but also the entire

embodied being, including physiological and sensory responses. It helps in recognizing that surveillance is not a purely cognitive phenomenon, but one that is deeply entwined with the physical and environmental aspects of an individual's experience. This insight can inform discussions on the ethical implications, psychological impacts, and potential interventions related to surveillance practices.

Thank you for this suggestion. Although we discuss previous work from social psychology that examines the impact of human surveillance on perceived autonomy, we agree that delving into the broader philosophical discourse surrounding surveillance and how it impacts agency is very interesting and would be a great avenue for future research. Due to space constraints, we do not delve into an extensive review of this literature, but we now briefly reference this important body of work in the introduction (please see p. 1) and encourage future research to do so in the general discussion (please see p. 30).

(5) Ethical Considerations: It would be crucial to highlight any steps taken to ensure participant well-being, privacy, and informed consent, especially in the context of a study involving surveillance.

We agree that it is crucial to ensure participant well-being in any study. We have highlighted steps taken to ensure such incredibly important ethical considerations in the “Overview of Studies” section (please see p. 10). For example, we note the following: “ This research was conducted in accordance with established ethical guidelines (e.g., informed consent was established, participant data was de-identified and stored on a secure server, and all participants were debriefed) and was approved by the Institutional Review Board at the authors’ university” (p. 10).

(6) Ecological Validity: Mentioning how the study's controlled settings align with or deviate from real-world surveillance scenarios would be valuable.

Thank you for this suggestion. We have added information about this to the general discussion, acknowledging the current limits of the present studies in terms of ecological validity and calling for future research to examine the observed effects in settings that would strengthen the ecological validity of the current findings. Specifically, we have included the following:

First, while the nature of our experimental designs enabled us to test the causal effects of surveillance source in controlled settings, future research should be conducted within more realistic organizational contexts to strengthen the ecological validity of the present findings. Future research in an organizational setting would also be better positioned to capture a greater range of resistance behaviors, along with the long-term effects of algorithmic (v. human) surveillance (p. 29).

(7) Long-term Effects of Surveillance: Considering the lasting impact of surveillance could provide a more comprehensive understanding of its psychological consequences.

We also agree that it is important to consider the long-term impacts of algorithmic surveillance as this would provide a more comprehensive understanding of its psychological consequences.

As shown above, we now call for future research to examine the effects of algorithmic surveillance over time (please see p. 29).

(8) Alternative Explanations: Acknowledging potential alternative factors that could influence participant responses would strengthen the study's validity.

Thank you for this suggestion. We now acknowledge, more extensively and in greater detail, alternative explanations for the observed effect (please see p. 30). We now discuss how, in addition to reduced perceived autonomy, “people might resist algorithmic surveillance because they feel uneasy about it due to its coldness (Bigman & Gray, 2018; Watyz et al., 2010; Young & Monroe, 2019; Zhang et al., 2022; but also see Yam et al., 2022), or because they have trouble concentrating with subjected to it (Unsworth & McMillan, 2013)” (p. 30). And we call for future work to consider these and other additional explanations.

(9) Technological Advancements: Considering the evolving nature of technology, it would be important to address how the study's findings might be influenced by future technological developments.

We agree that it is important to consider how the evolving nature of technology may impact individuals' perceptions of algorithmic (v. human) surveillance. We now discuss how future research should consider and continually reassess how the current findings may be influenced by technological developments and evolving societal discourse. Specifically, we included the following:

Moreover, considering the ever-evolving landscape of technology, it is imperative for future research to continually replicate, reassess, and analyze the influence of future technological developments on individuals' perceptions of autonomy and engagement in resistance to algorithmic (v. human) surveillance. As technology continues to evolve, the nature and capabilities of algorithmic surveillance are likely to become more sophisticated, which may lead to a shift in how individuals perceive and react to algorithmic surveillance. Relatedly, as the public discourse surrounding these technological developments continues to evolve, individuals' perceptions of and reactions to algorithmic (v. human) surveillance may also undergo significant changes. Future research should consider how these dynamic factors—technological developments and evolving societal discourse—might interplay and reshape how individuals perceive and react to algorithmic surveillance. (p. 31).

Response to Reviewer 2

The paper considers an important and timely research question, which is the impact of algorithmic surveillance on human behavior. Through four different studies, the authors provide convincing evidence this increasingly prevalent form of surveillance has a significant effect that should not be ignored in the age of AI. It is interesting to see that that

the effect is negated when framing the purpose of the surveillance as informational (as opposed to evaluative), which has practical implications. The paper is well written and easy to follow, and the literature is well covered.

Thank you so much. We are thrilled that you agree this is an important and timely research question. We are also elated that you found the evidence to be compelling, the review of the literature to be thorough, and the paper to be well-written. Thank you!

When I read the details of Study 2, I thought of a number of limitations due to differences that could have been avoided between the control and treatment conditions. However, I was pleased to see that the authors concluded their description of Study 2 by listing a number of limitations, which almost exactly matched my concerns. Later on, the authors presented Study 3, which adequately addressed all of my concerns regarding Study 2.

We also shared your concerns about the limitations of Study 2, and we are thrilled that you were pleased to see us list and then address the limitations in Study 3.

Minor comment:

In Study 1, I expected participants to be asked whether they recall an incident of being monitored (either by humans or by algorithms) before being asked about how they felt. This is because there could be many people who cannot even recall a single incident where they were monitored and evaluated by “artificial intelligence technology”. For such people, the task probably becomes more like a hypothetical question, e.g., if you were to imagine a situation where you are being monitored by an algorithm, how would you feel? Rather than: in a previous incident where you actually were monitored by an algorithm, how did you feel? Still, I don’t consider this to be a major comment, since the paper as a whole manages to provide sufficient evidence to support their claim.

Overall, I recommend the paper for publication.

We very much agree with your concern, and we tried to address it when we designed the study. We included two pre-screening attention check questions to ensure that participants could recall an incident in which they were monitored by a human and an incident in which they were monitored by an algorithm to gain access to the study. We also had participants describe the incidents before asking how they felt so that we could ensure participants recalled such events and exclude any participants who failed to do so (as pre-registered). We took these steps because we very much agree that in lieu of these steps, the task would likely become more like a hypothetical question. As you note, we did not explain this in the initial submission, and we have now added more detail regarding the study design and recruitment of participants to ensure greater clarity, and we believe the paper has improved as a result.

9th Feb 24

Dear Ms Schlund,

Your manuscript titled "I know I'm Being Watched: Algorithmic (v. Human) Surveillance Leads to Lower Perceptions of Autonomy and Increased Resistance" has now been seen by our reviewers, whose comments appear below. In light of their advice I am delighted to say that we are happy, in principle, to publish a suitably revised version in *Communications Psychology* under the open access CC BY license (Creative Commons Attribution v4.0 International License).

We therefore invite you to revise your paper one last time to address the remaining concerns of our reviewers and a list of editorial requests. At the same time we ask that you edit your manuscript to comply with our format requirements and to maximise the accessibility and therefore the impact of your work.

EDITORIAL REQUESTS:

SUBMISSION INFORMATION:

OPEN ACCESS:

Communications Psychology is a fully open access journal. Articles are made freely accessible on publication under a CC BY license (Creative Commons Attribution 4.0 International License). This license allows maximum dissemination and re-use of open access materials and is preferred by many research funding bodies.

For further information about article processing charges, open access funding, and advice and support from Nature Research, please visit <https://www.nature.com/commspsychol/article-processing-charges>

At acceptance, you will be provided with instructions for completing this CC BY license on behalf of all authors. This grants us the necessary permissions to publish your paper. Additionally, you will be asked to declare that all required third party permissions have been obtained, and to provide billing information in order to pay the article-processing charge (APC).

* **TRANSPARENT PEER REVIEW:** *Communications Psychology* uses a transparent peer review system.

On author request, confidential information and data can be removed from the published reviewer reports and rebuttal letters prior to publication. If you are concerned about the release of confidential data, please let us know specifically what information you would like to have removed. Please note that we cannot incorporate redactions for any other reasons.

* CODE AVAILABILITY: All Communications Psychology manuscripts must include a section titled "Code Availability" at the end of the methods section. We require that the custom analysis code supporting your conclusions is made available in a publicly accessible repository at this stage; please choose a repository that generates a digital object identifier (DOI) for the code; the link to the repository and the DOI must be included in the Code Availability statement. Publication as Supplementary Information will not suffice.

* DATA AVAILABILITY:

[link redacted]

Best regards,

Jennifer Bellingtier

Jennifer Bellingtier, PhD
Senior Editor
Communications Psychology

and on behalf of

Fernando Marmolejo-Ramos, PhD
Editorial Board Member
Communications Psychology
orcid.org/0000-0003-4680-1287

REVIEWERS' EXPERTISE:

Reviewer #1: AI alignment, AI in governance, philosophy of AI.

Reviewer #2: AI and human interactions, algorithms, data, and AI

REVIEWERS' COMMENTS:

Reviewer #1 (Remarks to the Author):

I am satisfied with the revision. Accept for publication.

Minor comments:

"Future research should evaluate the current findings in a variety of different samples, especially non-WEIRD" - you may want to word in in a more inclusive and sensitive manner, i.e. in a more diverse and representative sample that is not dominated by WEIRD features.

In point 6. the reader may benefit from detail in respect to what exactly would be required for a more realistic setting, and limitations and generalisations of the study should reflect this awareness.

On point 7. "or because they have trouble concentrating with subjected to it" something seems off with this sentence.

Reviewer #2 (Remarks to the Author):

I originally had only minor comments regarding the paper, and I believe the new version of the manuscript addresses these adequately. Based on this, I recommend the paper for publication.

Dear Members of the Review Team,

Thank you for your thoughtful feedback and for helping us improve the manuscript. We greatly appreciate the time and effort you have put into reviewing our work.

We believe we have addressed the issues raised. For your convenience, we have reproduced your comments in bold and have responded to each point below the reproduced text.

Very best,
Manuscript Authors

Response to Reviewer 1

I am satisfied with the revision. Accept for publication.

Thank you for helping us improve the manuscript!

Minor comments:

(1) "Future research should evaluate the current findings in a variety of different samples, especially non-WEIRD" - you may want to word in in a more inclusive and sensitive manner, i.e. in a more diverse and representative sample that is not dominated by WEIRD features.

Thank you for making this suggestion – we fully agree and have made the suggested change. Please see below:

“Future research should evaluate the current findings in a variety of different samples, especially in more diverse and representative samples.” (p. 28).

(2) In point 6. the reader may benefit from detail in respect to what exactly would be required for a more realistic setting, and limitations and generalisations of the study should reflect this awareness.

Thank you for making this suggestion—we agree that future research should test the effects in additional settings beyond the ones examined in this manuscript. We note the following:

“First, while the nature of our experimental designs enabled us to test the causal effects of surveillance sources in controlled settings, future research should be conducted within more realistic organizational contexts to strengthen the ecological validity of the present findings. Although we were able to assess real-time surveillance in a controlled lab setting, future research should assess real-time surveillance in an organizational setting where there are meaningful rewards and punishments for behavior. Future research in an organizational setting would also be better positioned to capture a greater range of

resistance behaviors, along with the long-term effects of algorithmic (v. human) surveillance.” (p. 27).

(3) On point 7. "or because they have trouble concentrating with subjected to it" something seems off with this sentence.

Thank you. We have corrected this sentence!

Response to Reviewer 2

I originally had only minor comments regarding the paper, and I believe the new version of the manuscript addresses these adequately. Based on this, I recommend the paper for publication.

Thank you so much for helping us improve the manuscript!